# Ultrastructure of the Sensilla on Antennae and Mouthparts of Larval and Adult *Cylas formicarius* (Coleoptera: Brentidae)

**DOI:** 10.3390/insects16030235

**Published:** 2025-02-21

**Authors:** Yuanchang Xu, Pengbo He, Faxu Lu, Mengjiao Li, Shahzad Munir, Mingfu Zhao, Yixin Wu, Yueqiu He, Guowen Tang

**Affiliations:** State Key Laboratory for Conservation and Utilization of Bio-Resources in Yunnan, College of Plant Protection, Yunnan Agricultural University, Kunming 650201, China; yuanchang1125@163.com (Y.X.); pengbohe@126.com (P.H.); 15911914162@163.com (F.L.); 18214015011@163.com (M.L.); shahzad_munir@ynau.edu.cn (S.M.); zhaomingfu@163.com (M.Z.); wyx680705@163.com (Y.W.); ynfh2007@163.com (Y.H.)

**Keywords:** *Cylas formicarius*, larva, adult, antennae, sensilla, scanning electron microscopy

## Abstract

*Cylas formicarius* is a significant agricultural pest worldwide. This research examined the fine structure of the heads of first- to third-instar larvae, as well as male and female adults, using scanning electron microscopy. The study particularly focused on the sensilla located on the mouthparts and antennae. Based on these observations, structural adaptations of the head and the potential functions of the sensilla were discussed. This information provides valuable insight into the developmental progression of the mouthparts and antennae sensilla of *Cylas formicarius*, aiding in the understanding of the species’ behavior.

## 1. Introduction

*Cylas formicarius* (Coleoptera: Brentidae) is a major pest of sweet potato (*Ipomoea batata* L.) and a recognized global quarantine threat [1]. This pest causes substantial damage to both the quality and yield of sweet potatoes [2,3]. Adult beetles feed on various parts of the plant and lay eggs in puncture wounds on the storage root epidermis [4]. The larvae tunnel into the roots and spend the entirety of their development within the storage roots [5], resulting in infestation during planting and storage [1,6]. Of the 109 countries worldwide that cultivate sweet potatoes [7], over 80 face production limitations due to *C. formicarius* [8]. In quarantine zones, damage to sweet potato roots by this pest ranges from 30% to 100%, and the affected area continues to grow annually, significantly impacting the sweet potato industry [9]. Currently, chemical insecticides are the main method of controlling *C. formicarius* [10], but prolonged pesticide use has led to resistance, making control increasingly difficult and complex [9,11]. Therefore, there is an urgent need for alternative sustainable methods to control the pest and manage its population growth. Using sex pheromones or food attractants for trapping is an environmentally friendly technique that prevents the development of resistance and is also an effective way to monitor pest populations [12,13]. Studying the types, distribution, and structure of sensilla plays a key role in understanding the feeding behaviors and mechanisms of insects, which is important for developing green pest control strategies [14].

Sensilla, specialized structures originating from the insect integument, are essential components of the sensory system [15]. These structures function as the primary receptors for sensory input, regulating various behaviors through the nervous system [16,17]. Sensilla are crucial for behaviors, such as feeding, defense, and reproduction [18]. The insect head is home to a diverse range of sensilla types, directly influencing the insect’s behavior. Thus, examining the types, morphology, and distribution of sensilla on the head is vital for understanding insect behavior. While the sensilla of larval and adult phytophagous Coleoptera have been extensively studied using scanning electron microscopy (SEM) in several families, such as Curculionidae [19,20], Scolytinae [21,22], Cerambycidae [23,24], Scarabaeidae [25], and Bruchidae [14,26], research on the ultrastructure of Brentidae is limited, and no studies have been conducted on the sensilla of *C. formicarius* larvae. Chen et al. [27] investigated the sensilla of adult *C. formicarius*, focusing on the antennae and maxillary palps; however, this study did not separate male and female samples and provided incomplete morphological data with unclear images. Therefore, it is necessary to refine these findings to better understand the behavioral responses of *C. formicarius*.

To investigate the behavioral mechanisms of *C. formicarius* larvae and adults, a comprehensive study was conducted to analyze the types, distribution, and morphology of sensilla on the heads of both stages using scanning electron microscopy (SEM). These results enhance the understanding of the development of the olfactory system in *C. formicarius* and provide critical insights for the development of pest control technologies and a deeper understanding of their behavioral ecology.

## 2. Materials and Methods

### 2.1. Insect Collection

*C. formicarius* specimens were collected from Wenshan Prefecture, Yunnan Province, China, on 8 October 2023. The adult specimens were stored in EP tubes containing 75% ethanol at 4 °C for future analysis. At the same time, storage roots with larvae were collected in the field, placed in Ziplock bags, and transported to the laboratory. These roots were kept in an MGC-300A illuminated thermostatic incubator, set to 27 ± 1 °C with 70 ± 5% humidity and an 8L/16D light/dark photoperiod.

### 2.2. Scanning Electron Microscopy (SEM)

A total of 10 adults of each sex and 10 larvae from the first, second, and third instars were randomly selected from the specimens and the storage roots. The samples were washed three times for 30 s each using an ultrasonic cleaner (SB-50, Scientz, Ningbo, China). Afterward, the heads were dissected under a stereomicroscope (KL200LED, Leica, Germany) and placed in a 4% glutaraldehyde solution at 4 °C for 12 h. The samples were then rinsed three times with 75% phosphate-buffered saline (PBS, 0.1 M, pH 7.2) for 10 min each. Gradient dehydration was performed with ethanol (30%, 50%, 70%, 90%, and 100% concentrations, with 20 min at each step), followed by drying at room temperature. The specimens were mounted on metallic stubs with conductive adhesive and sputter-coated with gold (E-1010, Hitachi, Tokyo, Japan) for 10 s. They were observed and photographed with a FlexSEM-1000 SEM (Hitachi, Japan) at accelerating voltages between 3 kV and 5 kV.

### 2.3. Terminology and Data Analysis

The terminology for head morphology and sensilla types adhered to the nomenclature established by Schneider [28]. Sensilla subtypes were classified according to the studies conducted by other researchers [14,20,29,30,31].

Head and sensilla parameters were measured using SEM particle-size statistics software (version: Hitachi Map 3D 3.0). Data analysis was performed using the SPSS Statistics 27.0 package. The differences in head length and width among the first, second, and third instar larvae, as well as the length, basal diameter, and quantity of mouthpart sensilla, were assessed using one-way analysis. The lengths and widths of the antennal segments, sensilla lengths, basal diameter of sensilla, and sensilla quantity from both male and female adults were compared using an independent samples *t*-test. A significance level of 0.05 was set for all data comparisons, with all results presented as mean ± standard error of the mean. Figures were generated using Origin 2024 software (version: OriginPro 2024b 10.1.5.132).

## 3. Results

### 3.1. Morphology of the Larvae’s Head and Sensilla

#### 3.1.1. Gross Morphology of the Larvae Head

The head of *C. formicarius* larvae is an elliptical capsule, with a notable increase in size across instars (Table 1, *p* < 0.05). The head surface is smooth, characterized by an inverted Y-shaped ecdysial cleavage line (Figure 1A). The head is symmetrically equipped with two types of setae: short and long (Figure 1B). The number of short setae remains constant at 18 across all instars, while the long setae count increases to 20, 24, and 26 in the 1st, 2nd, and 3rd instars, respectively. The chewing mouthparts are oriented ventrally and are hypognathous, consisting of the labrum, mandibles, maxillae, labium, and hypopharynx (Figure 1C). The labrum is inverted triangular with a deep clypeofrontal sulcus and contains four subtypes of sensilla chaetica. The mandibles are heavily ossified, with both a molar and incisor lobe, facilitating the effective cutting and grinding of food. The maxilla includes the cardo, stipes, galea, lacinia, and a maxillary palp. The maxilla and labium are interconnected and capable of coordinated movement. The hypopharynx is a hardened structure located centrally in the mouthparts, covered densely with spicules. The antennae, positioned behind the mandibles, are cone-shaped with a smooth outer surface and a blunt, rounded apex. They range in length from 14–19 μm and in diameter from 10–16 μm, with 7–10 conical sensilla distributed around them (Figure 1D). The size and number of these sensilla increase with instar. The lateral ocelli have degenerated and are no longer detectable.

#### 3.1.2. Types, Distribution and Morphology of Mouthparts Sensilla of *C. formicarius* Larvae

##### Sensilla Chaetica (SC)

SC are long, pointed, and present in large numbers. Four subtypes were identified, with the size of all subtypes increasing as the larvae progress through instars.

SC1 are found exclusively on the labrum, with four sensilla in total. They arise from a flexible circular socket and gradually taper to a pointed, slightly curved tip with a smooth outer surface (Figure 1E).

SC2 are short, with a smooth surface and slightly sharp tip (Figure 1E–G). These sensilla are present on the labrum, mandible, and maxilla, totaling 14 in number.

SC3 are symmetrically distributed anteriorly on the labrum, with two on each side. These sensilla are flattened in the lower and middle parts, with an extremely sharp, sword-like tip (Figure 1E). The size of SC3 increases with instar, with the third instar showing the largest and widest sensilla (Table 2).

SC4 are located on the inner side of the labrum and maxilla. They are shorter than SC1 and have a larger basal diameter. Their middle and upper sections are either curved or upright, with a smooth outer wall and flattened spine-like shape (Figure 1E,G). The number of SC4 sensilla was significantly higher than the others, with about 48 in total (Table 2).

##### Sensilla Basicaonica (SB)

Different regions contain distinct subtypes of maxilla and labium sensilla, categorized into two subtypes based on their length and morphology.

SB1 are found on the upper part of the terminal segment of the maxillary and labial palps. These sensilla are short, small, and cone-shaped with a smooth surface (Figure 1H,I). SB1 are the smallest receptors in the larval mouthparts, and their length and basal diameter increase significantly from the 1st to 3rd instars (Table 2).

SB2 are arranged upright on the labium, with only four present. In comparison to SB1, SB2 are longer and thicker (Figure 1H).

##### Sensillum Digitiformium (SD)

SD are positioned along the sides of each maxillary palp. They have a finger-like shape, lack a distinct pore structure, and are separated from the cuticle (Figure 1I). SD maintain a consistent shape, with lengths ranging from 12 to 18 μm and basal diameters between 3 and 6 μm (Table 2).

### 3.2. Morphology of Head and Mouthpart Sensilla of C. formicarius Adults

#### 3.2.1. Morphology of the Adults Head

The adult *C. formicarius* has a trunk-like head with a rostrum extending forward from the frontal area (Figure 2A,B). The rostrum of the male is significantly larger than that of the female, both in length (1510.00 ± 10.00 μm for males; 1442.33 ± 9.29 μm for females) and diameter (367.67 ± 4.51 μm for males; 348.00 ± 7.55 μm for females). The compound eyes protrude in an arch from the sides of the rostrum, covering about one-fourth of the total head surface (Figure 2A,B). Both male and female adults possess chewing-type mouthparts made up of five segments: a labrum, a pair of mandibles, a pair of maxillae, a labium, and a hypopharynx, forming the pre-oral cavity (Figure 2C). The head is covered with setae that are hair-like with fine tips, inserted in oval sockets at their base, and tilted about 30° towards the rostrum apex (Figure 2D). These setae are spread across the cuticle of the rostrum. In both males and females, the dorsal side has a noticeably higher density of setae compared to the ventral side, with males having significantly more setae (334.00 ± 9.17) than females (291.33 ± 8.50). The antennae are located between the mouthparts and the compound eyes, consisting of three parts: a scape, a pedicel, and a flagellum, which is composed of eight flagellomeres (F1–F8).

#### 3.2.2. Adult Rostral Sensilla Types, Distribution and Morphological Characteristics

ST is curved and covered with hairs, occurring in low numbers, and is classified into two subtypes.

ST1 are considerably longer than other sensilla types, with a bearded appearance, a shallow groove along the surface, and a tapered tip. These sensilla are arranged symmetrically in pairs within specialized “8”-shaped basal fossae, located in depressions on both sides of the labrum and labium (Figure 2D,F,G). In male *C. formicarius*, ST1 are significantly longer (128.33 ± 7.77 μm) and wider (4.89 ± 0.12 μm) than in females (Table 3).

ST2 are short, slender hairs with smooth surfaces (Figure 2D,E). This subtype is located only at the base of the maxilla. Their structure is similar to the setae found on the rostrum, except that ST2 branches into 2 to 3 parts at the base, specifically at the 1/5 position. No significant differences are found between males and females regarding ST2 (Table 3).

##### SC

SCs are the most common and abundant type of sensilla found on the mandible and maxilla and are divided into three subtypes.

SC1 have a rounded socket and taper towards the apex. Their surface is smooth, and their shape is acicular and aristate, measuring 37 to 39 μm in length (Figure 2D), making them the longest of the sensilla chaetica (Table 3).

SC2 are similar to SC1 but are shorter, with a smaller base diameter and a shallower socket (Figure 2D). They are positioned at an angle of about 45° to the cuticle (Table 3), and occur in limited numbers, with one on each mandible and two on each maxilla.

SC3 appear as spear-like structures and are much thicker than both SC1 and SC2. SC3 is predominantly found on the maxillary cuticle. They stand upright with a sharp tip, lack pores, and have a smooth surface (Figure 2F). The total number of SC3 is around 54, with no significant difference between males and females (Table 3).

##### SB

SB are tiny, cone-shaped sensilla with a blunt tip, and their apex is pierced by a pore (Figure 2G). These sensilla are found on the terminal segment of the maxillary (44.00 ± 0.00) and labial (20.00 ± 0.00) palps in both male and female *C. formicarius*. The SB are positioned at a 90° angle to the epidermis and have a length of less than 3 μm, making them the smallest sensilla present on the adult mouthparts (Table 3).

##### SD

SD are positioned on the lateral aspect of each maxillary palp. These sensilla have a consistent shape, with lengths ranging from 13 to 14 μm and widths between 2.4 and 2.6 μm. SD consist of a peg embedded in a cuticular groove and exhibit a finger-like form (Figure 2F,G). No differences are observed in the measurements of this sensilla between males and females (Table 3).

### 3.3. Comparison of the Distribution of Sensilla on the Mouthparts Between Larvae and Adults of C. formicarius

After the larvae matured into adults, both the types and quantities of sensilla on the mouthparts increased to varying degrees (Figure 3A,B). The number of sensilla types grew by one (larvae had three types and five subtypes, while adults had four types and five subtypes). The total number of mouthpart sensilla in adults was clearly higher, with a significant increase in SCs on the medial side of the mandibles compared to the larvae (adults 54.67 ± 1.15, larvae 36.33 ± 0.58). Furthermore, there was a larger number of SB on the maxillary palp (44.00 ± 0.00) and a reduced number on the labial palp (20.00 ± 0.00) in adults compared to larvae (maxillary palp 18.00 ± 0.00; labial palp 20.00 ± 0.00). SD showed the most considerable increase, with a threefold rise from larvae (one on each maxillary palp) to adults (three on each maxillary palp).

### 3.4. Morphology of Tentacle and Tentacle Sensilla of C. formicarius Adult

#### 3.4.1. Antennal Morphology

The antennae of both male and female *C. formicarius* adults are geniculate and comprise three segments: a scape, a pedicel, and eight flagellomeres (F1–F8). The scape is relatively thick at one end and slender at the other, connecting to the head via its slender end. The pedicel and the first to seventh subsegments of the flagellum are short and columnar, being noticeably shorter than the scape and the eighth flagellomere. A notable distinction exists between males and females in the shape of the eighth flagellomere. The eighth flagellomere in females exhibits a characteristic capitate shape (Figure 4A), whereas in males it is clavate (Figure 4B).

The average total length of the antennae is 1853.37 ± 7.86 µm for males and 1433.28 ± 7.91 µm for females, with the male antennae being significantly longer. The scape and eighth flagellomere are longer in males, while the pedicel and the first to eighth flagellomeres are shorter. Regarding width, the male antennae are wider at the first, fifth, and sixth subsegments of the flagellum, whereas the pedicel and eighth flagellomere are wider in females, with no significant difference observed in the width of the other subsegments (Figure 5A,B). A large number of cuticular pores are located at the base of the sensilla across all antennal segments. These pores are either round or oval in shape and occur in high abundance, although females display a significantly lower average pore count (575.67 ± 6.66) compared to males (672.33 ± 8.50).

#### 3.4.2. Types, Distribution, and Morphological Characteristics of Tentacle Sensilla of *C. formicarius*

##### ST

ST are hair-like protruding sensilla characterized by longitudinal grooves or smooth surfaces, primarily distributed on the eighth flagellomere (F8). These sensilla are further divided into two subtypes based on their length and shape: ST1 and ST2. Among these, ST1 exhibit marked sexual dimorphism in form, size, and quantity.

In females, ST1 is embedded in a tight, ridge-shaped socket, tapering towards the apex with blunt tips. These sensilla feature distinct crooked and deeply grooved surfaces and are curved in an arc-like shape at the middle (Figure 4C). The socket of male ST1 is similar to that of females; however, the outer wall of male ST1 is smooth, and a bulbous structure is present at the sensorium’s end (Figure 4D). The length, basal diameter, and number of ST1 are significantly greater in males than in females (Table 4). Furthermore, ST1 represents the largest and most abundant sensilla type on both male and female antennae.

ST2 are widely distributed across all antennal segments. Unlike ST1, the number of ST2 in females increases progressively from the third flagellomere to the distal flagellomere, although its density remains significantly lower than that of ST1. Structurally, ST2 resemble arc-shaped hairs that insert into antennal sockets, with their diameter decreasing sharply at about three-quarters of their length towards the tip, maintaining a smooth surface throughout (Figure 4E). No sexual differences are observed in the length, basal diameter, or quantity of ST2 (Table 4).

##### SC

SC sensilla were exclusively observed on the club and the eighth flagellum. These sensilla are classified into two subtypes based on apical bifurcation: SC1 and SC2.

SC1 are distinctly identifiable, with their tips extending beyond all other sensilla. They are positioned at an angle of about 31° to 34° relative to the antennal surface and are characterized by thick, robust, and slightly curved hair-like structures with pronounced longitudinal grooves and blunt tips (Figure 4C). The number of SC1 on males is about double that on females (Table 4).

SC2 are similar to SC1 in both morphology and size, differing primarily in the bifurcated tips, where two branches of unequal lengths form a “Y”-shaped structure (Figure 4F). The quantity of SC2 is significantly greater in males compared to females, highlighting a pronounced sexual dimorphism in their distribution (Table 4).

##### SB

Sensilla basiconica (SB) are distributed across the eighth flagellomere, with their density gradually increasing from the middle toward the apex. The total number of SB is about 27, with a length of 10–11 μm and a basal diameter of about 2 μm. These sensilla possess a smaller pinch angle compared to other sensilla, and no sex-related differences are observed in any individual parameters. The outer walls of SB display longitudinal lines and contain olfactory pores, while their overall structure is spike-shaped at the upper center (Figure 4G).

##### Sensilla Rod-like (SR)

SR sensilla exhibit distinctive longitudinal grooves on their walls and serrated apexes, with two subtypes identified based on length and basal diameter.

SR1 are arranged in a circular pattern around the fifth to eighth flagellomeres, primarily near their ends. Their quantity increases progressively from flagellomere 5 to 7, while on the eighth flagellum, their distribution is random. SR1 are the longest among this sensilla type, characterized by pronounced vertical grooves and wide tips. They emerge from invagination pits in the cuticle, and when viewed laterally, they resemble a stick truncated at the top (Figure 4H).

SR2 are shorter and less numerous than SR1 but are similarly arranged in circular patterns on the flagellum in both sexes. Structurally and morphologically, SR2 are nearly identical to SR1, differing only in size (Figure 4H). SR2 are more abundant in females compared to males, with no variations observed in other parameters (Table 4).

##### Böhm Bristles (BB)

BB are located between the scape and pedicel and at the connection point of the scape and head in adult antennae. These are the shortest sensilla observed on the adult antennae and are classified into two subtypes, BB1 and BB2, based on their morphological and ultra-micro characteristics.

BB1 have an arch-like structure with their bases positioned in cavity-shaped, wide sockets. These sensilla possess smooth surfaces (Figure 4I), are angled at about 30° to the antennal surface, and number about eight (Table 4).

BB2 are slightly shorter than BB1 (Table 4) and bifurcate at two-thirds of their length into a “V” shape, with branches of uneven thickness (Figure 4I).

##### Sensilla Furcatea (SF)

SF sensilla are slightly curved and resemble an awl or hair in shape. They are distributed on the eighth flagellum and are categorized into two subtypes, SF1 and SF2, based on their morphological and size characteristics. Both subtypes are significantly more abundant in males than in females.

SF1 are slightly curved at the base, which is embedded within a raised socket. Their walls are smooth, lacking furrows, and they possess short, obtuse side branches located at the middle or lower section, forming an angle of about 45° (Figure 4J). This subtype is relatively scarce, with males averaging 7.33 ± 1.15 and females averaging 4.67 ± 0.58. SF1 demonstrate marked sexual dimorphism in both quantity and length (Table 4).

In contrast, SF2 are shorter and less numerous compared to SF1. SF2 feature a granular protuberance at the base and a stout lateral branch, about half the diameter of the main branch, with the two branches positioned in close proximity (Figure 4K). A cone-shaped structure emerges from the middle of the sensilla at the apex. The number of SF2 in males is significantly higher than in females (Table 4), although no differences are observed in other parameters.

##### Sensilla Ligulate (SL)

SL are located on the surface of the eighth flagellum. They are straight and positioned at an angle of around 40° relative to the surface. The SL do not possess small pores or grooves on their surfaces. These sensilla are situated in a raised socket, compressed at the apex to a blunt tip (Figure 4L). Males display a significantly greater number of sensilla than females, while no significant difference in basal diameter was found between the sexes (Table 4).

## 4. Discussion

The structure of the head of *C. formicarius* larvae is characterized by a smooth epidermis, absence of ocelli, and degeneration of the antenna. These features are consistent with those observed in other drill-boring insects, such as *Ips typographus* [22], *Ips subelongatus* [22], *Cryptorrhynchus lapathi* [32], *Eucryptorrhynchus scrobiculatus*, and *E. brandti* [33]. The morphology of the mouthparts, antenna, and capsule of *C. formicarius* suggests an evolutionary adaptation to its specialized habitat. The larvae tunnel through and feed within sweet potato storage roots, with the smooth head capsule reducing resistance during burrowing. Larvae remain in dark environments for extended periods and do not require light perception, leading to the gradual evolutionary disappearance of the monocular eye. The antennal structure of *C. formicarius* larvae is simpler compared to non-boring insects, such as *Spodoptera frugiperda* (antennae with three segments and five sensilla types) [34], *Plutella xylostella* (three segments and six sensilla types) [35], and *Colobopterus quadratus* (five segments and three sensilla types) [25]. This simplification likely reflects the reduced necessity for host searching, making the structure better suited for burrowing within roots [36]. The reduced investment in ocelli and antennae enhances the larvae’s adaptability.

Three types and seven subtypes of sensilla are present on the head of *C. formicarius* larvae, namely SC (four subtypes), SB (two subtypes), and SD, resembling those of insects, like *Ips typographus* [22], *Ips subelongatus* [29], and *Cryptorrhynchus lapathi* [32]. Although the types and numbers of antennal sensilla remain constant, sensilla size increases significantly in higher instars. As multiple larvae inhabit the same root, intense competition occurs during later stages [6], and larger sensilla in higher instars may improve environmental perception and facilitate advantageous behavioral responses [34]. SC, recognized as mechanosensitive sensilla in insects, play roles in position selection and movement [37]. Four SC subtypes identified on the mouthparts likely sense mechanical stimuli and assist in feeding and burrowing. SB, known as chemical sensilla [38], includes two subtypes. SB1, found more frequently on the maxillary palp than the labial palp, appears to be sensitive to chemical signals and is associated with taste sensation, while SB2, distributed on the labium, likely performs a similar function. SD sensilla, involved in sensing vibration, temperature, humidity, and odorants in Coleoptera [22,39], may assist in evaluating the physiological condition of sweet potato storage roots.

The head structure of *C. formicarius* adults is more complex than that of larvae, featuring well-developed compound eyes and antennae. This adaptation is likely associated with an enhanced capacity for environmental interaction. Adults exhibit peak activity during the evening, engaging in foraging, mate searching, mating, and oviposition. These behaviors necessitate the ability to process extensive information on light, temperature, humidity, host volatiles, and conspecifics, necessitating the development of sophisticated compound eyes and antennae to meet these requirements [35]. The adult head extends forward, positioning the mouthparts at the anterior end, with highly sclerotized mandibles and labrum designed for efficient cutting and penetration.

The mouthparts of *C. formicarius* adults contain four types and five subtypes of sensilla: ST (two subtypes), SC (three subtypes), SB, and SD. Compared to larvae, adults possess a greater variety and quantity of sensilla, likely reflecting differences in dietary and environmental demands. Among these, ST sensilla are the most prevalent and are common across many insects, performing functions, such as sensing mechanical pressure [40], recognizing sex pheromones [41], and identifying host plants [42]. The ST1 subtype, exceptionally long in adults, is hypothesized to detect both mechanical vibrations and chemical signals, while the shorter ST2 subtype primarily senses host volatiles [20]. SC sensilla function as mechano-sensors, aiding in the creation of feeding nests, which subsequently facilitate oviposition by adults [43]. SB sensilla, located on the mandibular and lower lip whiskers, serve a gustatory role by providing initial food assessment [31]. SD sensilla, similar in function to those in larvae [20,22], show a threefold increase in adults, indicating a significant enhancement in sensory perception.

The adult head features highly developed antennae and compound eyes, with the F8 flagellum of the antenna undergoing specialization. In males, the F8 segment is considerably longer, about twice the length of the females’, while the middle part of the F8 segment in females is significantly enlarged. These adaptations greatly expand the surface area of the antennal tip, allowing for the placement of more receptors, which enhances the organism’s ability to detect sensory information [43]. The antennae of *C. formicarius* exhibit distinct sexual dimorphism, with male antennae being notably longer than those of females. This increased length of male antennae may serve an adaptive function by providing a greater number of sensilla, which likely enhances the detection of pheromones. Chen et al. [27] identified four types of sensilla on the antennae of adult *C. formicarius*, including sensilla trichodea, rod-like sensilla, sensilla chaetica, and sensilla ampullacea. In this study, in addition to the three previously observed types, four new sensilla types were identified on the antennae of both male and female specimens: SB, BB, SF, and SL. Additionally, significant differences in sensilla subtypes were noted, with ten subtypes identified, including two subtypes each of ST, SC, SR, BB, and SF. This variation may be due to limitations in the previous SEM technique and differences in the terminology used to classify the sensilla [34].

The functions of the same sensilla can vary depending on their position, the developmental stage of the insect, or its sex. For example, ST located on the mouthparts generally act as contact receptors, whereas those on the antennae are more responsive to volatile compounds [44]. In various Coleoptera species, ST are involved in detecting mechanical and chemical signals [41]. In *C. formicarius* adults, ST1 exhibit clear sexual dimorphism in morphology, quantity, and size, indicating that their function differs between males and females. Males possess significantly more ST1 than females, while females show a greater number of notches. It is suggested that the increased number of ST1 in males may assist in detecting pheromonal signals, while the notches in females may aid in creating oviposition sites and receiving mechanical stimulation to ensure suitable conditions for egg-laying. ST2 are present in every segment of the antennae of adult *C. formicarius*. These are shorter and smaller than ST1, which suggests a reduced tactile function [16,17], and primarily serve to detect olfactory cues from host plants. Two subtypes of ST2 were identified in adults, with males showing a higher abundance than females. Since SC act as a mechanosensory organ [45], it is speculated that this may be linked to the male’s need to detect mechanical resistance from females during mating. SB are chemoreceptors with olfactory functions and can sense a variety of volatile chemicals [29]. The SB of adults form a unique ridge-cluster pattern, which may expand the area for odor detection, thus improving the efficiency of capturing odor molecules and enhancing olfactory and gustatory abilities. Zuo [46] proposed that SR may play a role in sensing female pheromones, as their number is greater in males. However, in females, the two subtypes of SR are significantly more numerous than in males, and these sensilla are long and susceptible to physical stimulation. This suggests that males may use SR to detect female pheromones, while females may rely on the physical stimuli from SRs to find suitable oviposition sites. The distribution of BB in this study is consistent with that observed in other weevils [47,48], being located between the scape and head, as well as at the junction of the scape and pedicel in *C. formicarius* adults. However, *C. formicarius* is unique in having two subtypes of BB, a rare feature in other weevils. BB functions as a gravity sensor, regulating the position and movement of the antennal organ [49]. The bifurcated structure of BB2 may suggest that its sensory capabilities are stronger than those of BB1. The SF found in Lepidoptera [34,50], and Hemiptera [51] differ significantly in morphology from the SF on the antennae of *C. formicarius*, which are longer and thicker. Studies indicate that SF function to sense both mechanical and chemical stimuli [52]. In this study, SF1 are longer than SF2, increasing their surface area and enhancing their olfactory function, indicating that SF1 may be more effective than SF2. Some researchers consider SF to be a specialized form of ST, performing similar functions [53]. SL, which is rare in Coleoptera, have an unclear function. The morphology of SL’s base and outer wall in this study resembles that of ST, suggesting that SL may also have tactile and olfactory functions. Further investigation is necessary to confirm the exact functions of these sensilla in *C. formicarius* larvae and adults.

## Figures and Tables

**Figure 1 insects-16-00235-f001:**
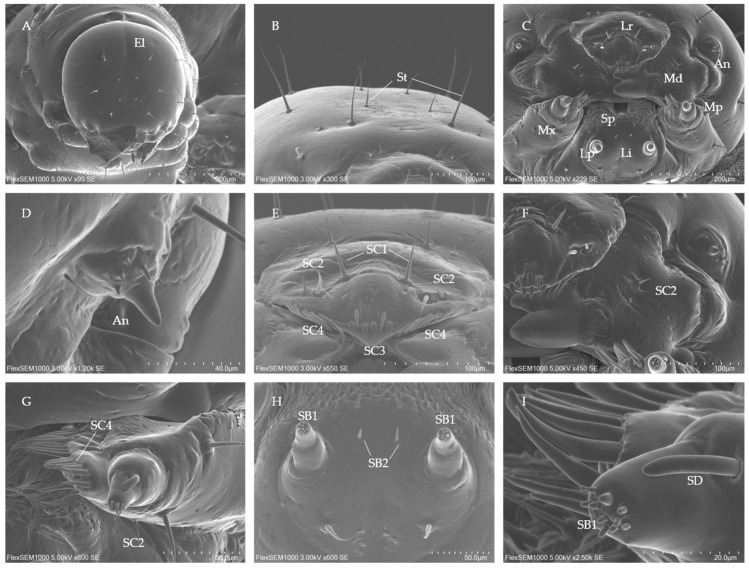
Head and sensilla morphology of *C. formicarius* larvae. (**A**) Dorsal view of the head. (**B**) Dorsal magnified view of the head. (**C**) Front view of the mouthparts. (**D**) Magnified view of the antennae. (**E**) Front view of the labrum. (**F**) Front view of the mandible. (**G**) Front view of the maxilla. (**H**) Front view of the labium. (**I**) Ventral view of the maxillary palp. An, antennae; Lp, labial palp; Lr, labrum; Li, labium; Md: mandible; Mp, maxillary palp; Mx, maxilla; Sp: Spicule; SB (1~2), sensilla basicaonica (1~2), SC (1~4); sensilla chaetica (1~4); SD, sensilla digitiformia.

**Figure 2 insects-16-00235-f002:**
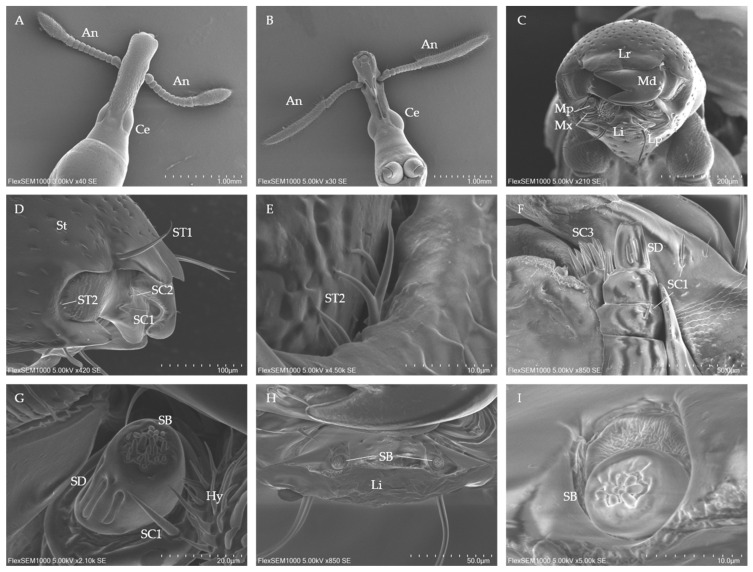
Morphology of the head and sensilla of adult *C. formicarius* beetles. (**A**) Dorsal view of the female head. (**B**) Ventral view of the male head. (**C**) Front view of the mouthparts. (**D**) Lateral view of the mouthparts. (**E**) Sensilla trichodea 3. (**F**) Lateral view of the maxilla. (**G**) Close-up view of the maxillary palp. (**H**) Overall view of the labium. (**I**) Anterior view of the apical labial palp. Ce, compound eye; An, antennae; Hy, hypopharynx; Li, labium; Lp, labial palp; Lr, labrum; Md, mandible; Mp, maxillary palp; Mx, maxilla; ST (1~2), sensilla trichodea (1~2); SB, sensilla basicaonica; SC (1~3), sensilla chaetica (1~3); SD: sensilla digitiformia.

**Figure 3 insects-16-00235-f003:**
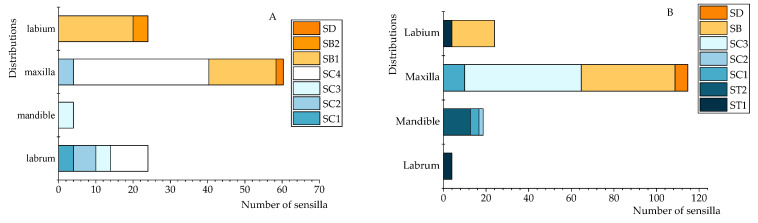
Distribution patterns of mouthpart sensilla in *C. formicarius* larvae and adults. (**A**) Larval mouthparts. (**B**) Adult mouthparts.

**Figure 4 insects-16-00235-f004:**
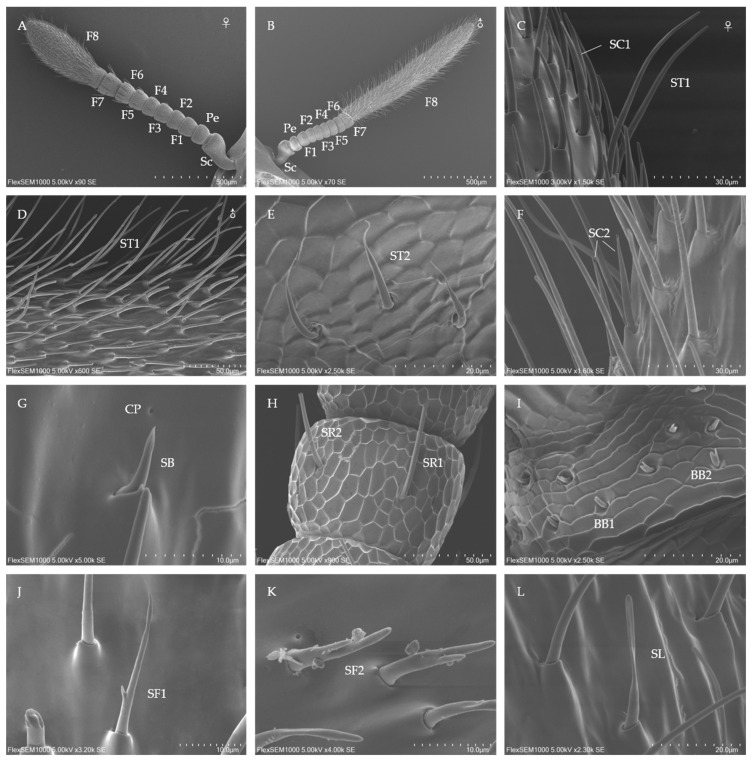
Morphological features of the antennae and sensilla in male and female adult *C. formicarius*. (**A**): female antennae; (**B**): male antennae; (**C**–**L**): antennae sensilla. Sc: scape; Pe: pedicel; F1~F8: first flagellum~eighth flagellum; ST (1~2): sensilla trichodea (1~2); SC1~2: sensilla chaetica (1~2); SB: sensilla basicaonica; CP: cuticular pore; SR (1~2): sensilla rod-like (1~2); BB (1~2): Böhm bristles 1~2; SF (1~2): sensilla furcatea (1~2); SL: sensilla ligulate.

**Figure 5 insects-16-00235-f005:**
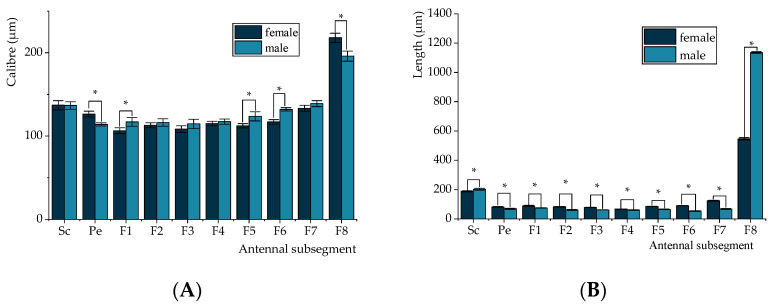
Measurements of antenna segment length and width in adult *C. formicarius*. (**A**) Antenna length. (**B**) Antenna width. Sc: scape; Pe: pedicel; F1~F8: first to eighth flagellum. Asterisks “*” denote significant differences between male and female adults for the same antenna subsegments (*p* < 0.05).

**Table 1 insects-16-00235-t001:** Head length and width of 1st to 3rd instar larvae of *C*. *formicarius*.

Instar	Length (μm)	Width (μm)
1st	643.00 ± 8.72 c	519.33 ± 8.39 c
2nd	803.00 ± 10.82 b	722.33 ± 10.12 b
3rd	1004.67 ± 11.02 a	904.33 ± 9.61 a

Data are mean ± SE, different letters indicate significant differences among instar larvae for length or width (*p* < 0.05).

**Table 2 insects-16-00235-t002:** Length, basal diameter, pinch angle, total number, and distribution position of head sensilla of the 1st to 3rd instar larvae of *C. formicarius*.

Type	Instar	Length (μm)	Base Diameter (μm)	Quantity	Distributions
SC1	1st	40.27 ± 0.93	3.94 ± 0.10	4.00 ± 0.00	Labrum
	2nd	44.53 ± 0.67	4.54 ± 0.09	4.00 ± 0.00	
	3rd	55.27 ± 2.22	4.82 ± 0.09	4.00 ± 0.00	
SC2	1st	8.61 ± 0.45	2.59 ± 0.18	14.00 ± 0.00	Labrum, mandible, maxilla
	2nd	14.63 ± 1.31	3.14 ± 0.10	14.00 ± 0.00	
	3rd	17.67 ± 1.31	3.42 ± 0.16	14.00 ± 0.00	
SC3	1st	13.80 ± 0.50	6.62 ± 0.54	4.00 ± 0.00	Labrum
	2nd	16.48 ± 0.99	7.67 ± 0.11	4.00 ± 0.00	
	3rd	22.61 ± 1.54	7.72 ± 0.20	4.00 ± 0.00	
SC4	1st	24.16 ± 0.65	3.94 ± 0.10	47.67 ± 0.57	Labrum, maxilla
	2nd	27.64 ± 0.88	4.54 ± 0.09	47.33 ± 1.15	
	3rd	34.31 ± 1.07	4.86 ± 0.08	48.33 ± 0.58	
SB1	1st	1.98 ± 0.03	1.62 ± 0.11	38.00 ± 0.00	Maxillary palp, labial palp
	2nd	2.36 ± 0.10	2.00 ± 0.12	38.00 ± 0.00	
	3rd	3.56 ± 0.12	2.67 ± 0.03	38.00 ± 0.00	
SB2	1st	10.25 ± 0.43	2.77 ± 0.17	4.00 ± 0.00	Labium
	2nd	11.05 ± 1.49	3.13 ± 0.09	4.00 ± 0.00	
	3rd	13.83 ± 0.65	3.61 ± 0.05	4.00 ± 0.00	
SD	1st	12.67 ± 0.87	3.48 ± 0.23	2.00 ± 0.00	Maxillary palp
	2nd	14.43 ± 0.71	4.19 ± 0.10	2.00 ± 0.00	
	3rd	17.73 ± 0.86	5.32 ± 0.13	2.00 ± 0.00	

Data are expressed as mean ± SE.

**Table 3 insects-16-00235-t003:** Length, basal diameter, angle, total number and distribution of oral sensilla of male and female adult *C. formicarius* beetles.

Types	Sex	Length (μm)	Base Diameter (μm)	Quantities	Distributions
ST1	Female	93.60 ± 6.17	4.26 ± 0.18	8.00 ± 0.00	Labrum, labium
	Male	128.33 ± 7.77 *	4.89 ± 0.12 *	8.00 ± 0.00	Labrum, labium
ST2	Female	17.53 ± 1.10	1.07 ± 0.07	14.67 ± 1.15	Mandible
	Male	16.70 ± 1.01	1.15 ± 0.07	15.33 ± 1.15	Mandible
SC1	Female	38.63 ± 2.86	2.26 ± 0.12	8.00 ± 0.00	Mandible, maxilla
	Male	37.70 ± 3.08	2.30 ± 0.11	8.00 ± 0.00	Mandible, maxilla
SC2	Female	14.63 ± 1.97	1.71 ± 0.24	6.00 ± 0.00	Mandible
	Male	15.43 ± 0.61	1.81 ± 0.17	6.00 ± 0.00	Mandible
SC3	Female	29.17 ± 1.59	3.65 ± 0.14	53.33 ± 1.15	Maxilla
	Male	30.67 ± 1.72	3.54 ± 0.07	54.00 ± 2.00	Maxilla
SB	Female	2.62 ± 0.63	1.46 ± 0.20	64.00 ± 0.00	Maxillary palp, labial palp
	Male	2.44 ± 0.12	1.59 ± 0.17	64.00 ± 0.00	Maxillary palp, labial palp
SD	Female	13.97 ± 1.23	2.49 ± 0.21	6.00 ± 0.00	Maxillary palp
	Male	14.40 ± 1.73	2.53 ± 0.09	6.00 ± 0.00	Maxillary palp

Data are mean ± SE. “*” indicates significant difference between male and female adults for the same sensilla type (*p* < 0.05).

**Table 4 insects-16-00235-t004:** Length, basal diameter, angle of pinch, total number, and distribution position of adult *C. formicarius* tentacle sensilla.

Type	Sex	Length (μm)	Base Diameter (μm)	Quantities	Distributions
ST1	Female	95.23 ± 4.27	3.84 ± 0.11	122.00 ± 7.00	F8
	Male	125.67 ± 6.66 *	4.32 ± 0.09 *	524.67 ± 8.14 *	F8
ST2	Female	31.23 ± 2.51	2.28 ± 0.15	31.67 ± 2.89	S, P, F1–F8
	Male	28.87 ± 1.68	2.20 ± 0.13	33.67 ± 2.08	S, P, F1–F8
SC1	Female	33.60 ± 3.18	3.29 ± 0.18	32.67 ± 3.06	F8
	Male	35.13 ± 3.46	3.21 ± 0.24	63.00 ± 3.61 *	F8
SC2	Female	32.77 ± 1.99	3.22 ± 0.10	21.00 ± 2.65	F8
	Male	34.53 ± 1.96	3.33 ± 0.16	33.67 ± 2.52 *	F8
SB	Female	11.21 ± 1.75	2.04 ± 0.12	27.00 ± 2.00	F8
	Male	10.68 ± 2.48	2.12 ± 0.25	26.67 ± 2.08	F8
SR1	Female	64.60 ± 2.03	4.12 ± 0.12	163.33 ± 4.51 *	F5–F8
	Male	61.93 ± 3.04	4.09 ± 0.15	97.00 ± 5.57	F5–F8
SR2	Female	41.73 ± 3.07	3.45 ± 0.11	69.67 ± 4.16 *	F1–F8
	Male	43.00 ± 1.95	3.57 ± 0.13	49.33 ± 3.79	F1–F8
BS1	Female	4.02 ± 0.20	1.72 ± 0.14	8.00 ± 1.00	Between the scape and pedicel, between the scape and head
	Male	4.06 ± 0.08	1.65 ± 0.07	8.33 ± 1.15	Between the scape and pedicel, between the scape and head
BS2	Female	3.95 ± 0.10	1.80 ± 0.07	10.00 ± 2.00	Between the scape and pedicel, between the scape and head
	Male	3.87 ± 0.17	1.66 ± 0.06	9.33 ± 3.21	Between the scape and pedicel, between the scape and head
SF1	Female	24.50 ± 1.06	1.92 ± 0.09	4.67 ± 0.58	F8
	Male	29.73 ± 2.46 *	2.06 ± 0.15	7.33 ± 1.15 *	F8
SF2	Female	16.00 ± 1.71	2.01 ± 0.16	3.33 ± 0.58	F8
	Male	17.10 ± 1.65	2.20 ± 0.13	6.00 ± 1.00 *	F8
SL	Female	33.13 ± 2.39	3.03 ± 0.09	10.67 ± 2.08	F8
	Male	33.70 ± 2.40	3.15 ± 0.17	18.33 ± 3.21 *	F8

Data are mean ± SE. “*” indicates significant difference between male and female adults for the same sensilla type by independent-samples *t*-test (*p* < 0.05).

## Data Availability

The data presented in this study are available on request from the corresponding author.

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
