# Peer review of "Ultrastructure of the Sensilla on Antennae and Mouthparts of Larval and Adult Cylas formicarius (Coleoptera: Brentidae)"

_insects, 2025, doi:10.3390/insects16030235_

Round 1

Reviewer 1 Report

Comments and Suggestions for Authors

This study attempted to fully reveal the head and antenna structures of both adults and larvae of C. formicarius. I believe that it is a valuable study in terms of systematic taxonomy. The study was supported by SEM images. The SEM image of the antennas clearly reveals sexual dimorphism. Whether these morphological structures are important taxonomic data in species identification was compared with some other species. This makes the study even more valuable. In accordance with the purpose of the study, it was also revealed with literature data how the sensilla located on the head and antennas direct the behavioral characteristics of the species.

In addition to all these situations, the following problems need to be fixed:

1. In line 17, a dot is placed after the genus name. It must be deleted. I think there was a spelling mistake.

2. The author contributions section should also state who made the species identification.

Author Response

Thank you for the opportunity to revise our manuscript titled “Ultrastructure of the sensilla on antennae and mouthparts of larval and adult Cylas formicarius (Coleoptera: Brentidae)” (insects-3459192). We sincerely appreciate the valuable feedback from you, which has significantly improved our work.

Below are our responses to the reviewers’ comments:

Comments 1: In line 17, a dot is placed after the genus name. It must be deleted. I think there was a spelling mistake.

Response 1: Thank you for the reminder. The revision based on the reviewer’s comment is as follows (Lines 16-17): “This information provides valuable insight into the developmental progression of the mouthparts and antennae sensilla of Cylas formicarius.

Comments 2: The author contributions section should also state who made the species identification.

Response 2: Thank you for the reminder. The revision based on the reviewer’s comment is as follows (Line 498): Author Contributions: Conceptualization, Y.X. and G.T.; species identification, G.T.

Reviewer 2 Report

Comments and Suggestions for Authors

The MS entitled Ultrastructure of the sensilla on antennae and mouthparts of 2 larval and adult Cylas formicarius (Coleoptera: Brentidae) by  Yuanchang Xu, Pengbo He, Faxu Lu, Mengjiao Li, Shahzad Munir, Mingfu Zhao, Yixin Wu, Yueqiu He and  Guowen Tang deals with the ultrastructural description of the sensilla of the head of this pest insect in the larval instars and adult stage (male and female).

The aim of the paper is a deeper understanding of the development of the olfatory system in Cylas formicarius. The function of the different types of sensilla were discussed, providing a theoretical basis for future studies on the behavioral ecology of this pest.

The article is well conceptualized and written. All the sections are accurately described. The SEM images reveal a technically good treatment of the samples. Data have been clearly presented and statistically analized. In the discussion, results have been well argumented and properly accompanied by literature citations.

Thus, I strongly suggest the paper has to be published in Insects.

Only a few minor comments, listed below

References 2 and 3 have not been cited. Please, introduce them in the text or remove them from the list of the literature in the reference section (renumbering all the references).

Line 67: Chen JX [27]. Change as Chen et al. [27]

Please, add the measure unit in the header line of the Tables 2 and 3 …Length (mm)…. Base diameter (mm)

Format appropriately the spacing in the header line of the Table 4….Length (mm) ……Base diameter (mm) and in the caption of the ordinate axis of the Fig. 5 A  Length (mm)  and 5B Calibre (mm) 

Author Response

Thank you for the opportunity to revise our manuscript titled “Ultrastructure of the sensilla on antennae and mouthparts of larval and adult Cylas formicarius (Coleoptera: Brentidae)” (insects-3459192). We sincerely appreciate the valuable feedback from you, which has significantly improved our work.

Below are our responses to the reviewers’ comments:

Comments 1: References 2 and 3 have not been cited. Please introduce them in the text or remove them from the list of the literature in the reference section (renumbering all the references).

Response 1: Thank you for the reminder. We sincerely apologize for omitting references 2 and 3 in the previous manuscript. These references should have been cited after the sentence: “This pest causes substantial damage to both the quality and yield of sweet potatoes.” The revision based on the reviewer’s comment is as follows (Lines 39-40): This pest causes substantial damage to both the quality and yield of sweet potatoes [2,3].

Comments 2: Line 67: Chen JX [27]. Change as Chen et al. [27].

Response 2: Thank you for the reminder. The revision based on the reviewer’s comment is as follows (Line 67): Chen et al [27] investigated the sensilla of adult C. formicarius.

Comments 3: Please, add the measure unit in the header line of the Tables 2 and 3 …Length (mm)…. Base diameter (mm).

Response 3: Thank you for the reminder. The revision based on the reviewer’s comment is as follows (Table 1, 2, 3 and 4): We will update the header rows of Tables 2, 3 and 4 to "Length (μm)" and "Width (μm)," and we will also revise the "Length/μm" sum in Table 1 to "Length (μm)" and "Width (μm)" in order to standardize the format.

Comments 4: Format appropriately the spacing in the header line of the Table 4….Length (mm) ……Base diameter (mm) and in the caption of the ordinate axis of the Fig. 5 A Length (mm) and 5B Calibre (mm).

Response 4: Thank you for the reminder. The revision based on the reviewer’s comment is as follows (Figures 5A and 5B): The ordinate title in Figure 5A will be updated from "Calibre(μm)" to "Calibre (μm)," and the ordinate title in Figure 5B will be updated from "length(μm)" to "Length (μm)" to ensure proper formatting.